# A Comprehensive Benchmarking and Systematic Analysis of Deep Learning Models for Sonomammogram Segmentation

**Malitha Gunawardhana**                    MALITHA.GUNAWARDHANA@AUCKLAND.AC.NZ
*Auckland Bioengineering Institute, University of Auckland, New Zealand*

**Norbert Zolek**                    NZOLEK@IPPT.PAN.PL
*Institute of Fundamental Technological Research of the Polish Academy of Sciences, Warsaw, Poland*

**Editors:** Accepted for publication at MIDL 2026

## Abstract

Accurate segmentation of breast lesions in sonomammograms supports computer assisted diagnosis and early breast cancer detection. Existing public ultrasound datasets contain duplicates, mislabeled cases, and non-breast images, which leads to unreliable model evaluation. To address this, we construct a curated multi-centre dataset of 3,494 images with expert-verified annotations and patient-level splits. Using this dataset, we define a unified benchmarking protocol and evaluate eleven representative architectures, including nnU Net variants, SegResNet, SwinUNETR, U Mamba, and SAMed. All models are trained and assessed under identical preprocessing, training, and evaluation settings. Performance is measured with Dice, Sensitivity, Specificity, Accuracy, and Hausdorff Distance metrics. We also analyse how loss function choice and training data volume influence performance. SAMed p512 obtains the best Dice score at $0.860 \pm 0.141$ and the lowest Hausdorff Distance at $3.896 \pm 5.472$. The benchmark provides a reproducible reference for breast ultrasound segmentation and clarifies how architecture design and data-related factors shape performance in this setting.

**Keywords:** Segmentation, Sonomammogram, Deep Learning, Benchmarking

## 1. Introduction

Breast cancer is the most frequently diagnosed malignancy among women worldwide and remains a major global health burden. In 2022, more than 2.3 million new cases and approximately 670,000 deaths were reported, underscoring the critical need for early detection and reliable diagnostic tools (Chaudhari et al., 2024; Gunawardhana and Zolek, 2025). Nearly half of all diagnosed cases occur in women without identifiable risk factors apart from age and sex, and although male breast cancer is uncommon, it accounts for roughly 0.5 to 1 percent of diagnoses (Tarannum, 2024). These observations highlight the importance of accurate and accessible imaging-based assessment.

Mammography is the clinical standard for breast cancer screening, yet its diagnostic performance is reduced in dense breast tissue, and it introduces exposure to ionising radiation. Ultrasound imaging offers an important complementary role because it is non-ionising, widely available, cost-effective, and suitable for repeated examinations (Dan et al., 2024; Guo et al., 2018; Sahu et al., 2024). It provides real-time visualisation of soft tissues and is commonly used to detect and characterise breast lesions in both diagnostic and screening workflows. However, ultrasound interpretation is operator dependent, image quality varies across devices and users, and artefacts can obscure lesion boundaries. These challenges

motivate computational methods that can provide consistent and reproducible guidance in clinical practice.

Deep learning has become central to medical image analysis, with convolutional neural networks enabling high performing models for detection, classification, and segmentation. Segmentation of breast lesions in ultrasound is particularly important because it defines lesion extent and supports the extraction of morphological descriptors relevant to diagnosis. U Net (Ronneberger et al., 2015) and its derivatives serve as widely adopted baselines, and more recent architectures introduce attention mechanisms, transformers, and state space modelling. Despite this growing landscape, performance comparisons across studies remain difficult to interpret.

The first obstacle to consistent progress is the lack of standardised benchmarking. Existing studies often evaluate only a limited set of models and differ in their choice of data splits, preprocessing procedures, and evaluation metrics. As a result, reported improvements are not directly comparable. Variations in experimental design can mask or exaggerate architectural differences, which limits the ability to assess real performance gains.

A second challenge concerns the integrity of widely used datasets. One of the most frequently cited ultrasound datasets of this type contains numerous errors and inaccuracies (Pawlowska et al., 2023; Żołek and Pawłowska, 2025) as identified duplicate images, inconsistent labels, and non-breast scans that remained in the collection. Some images also contain overlaid annotations within lesion regions. Without patient-level separation and integrity checks, such issues can introduce information leakage (Kuhn and Johnson, 2019) during random splitting, which leads to overly optimistic performance estimates. These findings highlight the importance of dataset validation when developing and evaluating segmentation models.

Reliable benchmarking requires both high-quality data and a unified evaluation protocol. Architectural advances are difficult to interpret in the absence of validated datasets and consistent experimental settings. Establishing a reproducible framework is therefore essential for determining how different models behave under controlled conditions and for providing reference points for future research.

This study addresses these gaps by introducing a curated sonomammogram dataset that has been screened for duplicates, label inconsistencies, and non-breast images. We use this dataset to evaluate eleven representative deep learning architectures for breast lesion segmentation. These include classical encoder-decoder networks, attention-enriched designs, transformer-based models, and state space architectures. All models are trained and evaluated within a unified pipeline that standardises preprocessing, train-test division, and metric computation. We assess performance using the Dice coefficient, sensitivity, specificity, accuracy, and boundary-based measures.

This work makes three main contributions. (1) We introduce a curated, multi-center dataset of 3,494 ultrasound images with expert-verified annotations and patient-level separation, along with explicit integrity checks for duplication and label consistency. Sample of this dataset consisting 256 cases is already published (Pawłowska et al., 2024) and the full dataset is prepared for similar publication. (2) We present a unified benchmarking protocol that evaluates eleven deep learning architectures under identical preprocessing, training, and evaluation conditions. (3) We perform systematic analyses of architectural

performance, loss function choice, and training data volume to characterise factors that influence segmentation accuracy and boundary quality.

Rather than proposing a new segmentation model, this study establishes a reproducible benchmark built on validated data and a transparent evaluation pipeline. The benchmark provides a reliable reference for future work in breast ultrasound segmentation and supports the development of methods grounded in verifiable data quality.

## 2. Methodology

### 2.1. Dataset

This study utilises a newly curated dataset comprising 3,494 two-dimensional sonomammogram images collected from 1,727 patients across multiple diagnostic centres in Poland between 2019 and 2024. Each image is paired with a corresponding binary segmentation mask delineating the tumor region. The annotations were created independently by five board-certified radiologists, each with a minimum of ten years of experience in breast imaging. Disagreements were resolved through consensus review, and all lesion-level diagnoses were pathologically confirmed through biopsy or validated via follow-up imaging to ensure diagnostic accuracy.

All images were standardised by resizing to $512 \times 512$ pixels while preserving anatomical proportions. Images depicting benign and malignant lesions were included; however, normal scans without visible tumours were excluded since they provide no relevant segmentation information. To prevent data leakage and ensure realistic model evaluation, the dataset was split at the patient level, such that all images from a single patient appeared exclusively in either the training or testing set. The split was fixed at 80% of patients for training and 20% for testing, ensuring strict separation between seen and unseen subjects.

This dataset represents a significant step toward addressing the limitations of prior public breast ultrasound datasets, which have been shown to contain duplication, labelling errors, and non-breast images (Pawlowska et al., 2023). By ensuring data integrity, expert consensus, and patient-level segregation, the dataset provides a reliable foundation for reproducible benchmarking and model comparison.

### 2.2. Model Architectures

To comprehensively assess the state of deep learning for sonomammogram segmentation, we benchmarked eleven segmentation architectures spanning conventional encoder–decoder designs, residual and attention-based extensions, and recent transformer and sequence-modelling variants. Each model was configured using publicly available implementations, with hyperparameters and training protocols standardised across experiments to ensure fairness.

The evaluated models include:

- nnU-Net and its variants (Isensee et al., 2024): a self-configuring U-Net framework that automatically adapts preprocessing, architecture depth, and training parameters to the dataset. Both standard and residual versions were evaluated.

- LightMUNet (Liao et al., 2024): a lightweight, multi-scale U-Net variant optimised for computational efficiency without significant performance degradation.

- SegResNet (Myronenko, 2019): an encoder–decoder model combining residual learning with U-Net topology, designed to capture deep semantic features while maintaining localisation precision.

- SwinUNETR (Hatamizadeh et al., 2021): a transformer-based U-Net utilising shifted-window attention mechanisms to capture long-range spatial dependencies, representing the transformer paradigm in medical segmentation.

- U-MambaBot and U-MambaEnc (Ma et al., 2024): models integrating Mamba sequence modelling blocks into the segmentation pipeline. The U-MambaBot architecture introduces a Mamba-based bottleneck module between the encoder and decoder, while U-MambaEnc replaces the encoder entirely with Mamba layers, enabling dynamic context propagation through learned state-space representations.

- SAMed (Zhang and Liu, 2023): a medical adaptation of the Segment Anything Model (SAM) fine-tuned for ultrasound data. Two configurations were tested with patch sizes of 256 and 512, to evaluate the trade-off between local and global context modelling.

This diverse selection encompasses the major design families in contemporary segmentation, including CNNs, Transformers, and sequence-based architectures, allowing for an unbiased and systematic comparison across paradigms.

## 2.3. Training and Evaluation

All models were implemented in PyTorch and trained on NVIDIA H100 PCIe GPUs. Training was conducted for 1,000 epochs with a batch size of 16. The Adam optimizer was used with an initial learning rate of 0.01, governed by a polynomial decay scheduler to ensure smooth convergence. Unless otherwise noted, the primary loss function was DiceCE, a hybrid of Dice loss and Cross-Entropy loss that balances region overlap with pixel-wise classification accuracy.

To ensure robust and unbiased evaluation, five-fold cross-validation was performed. Model performance was assessed on the test set and reported as mean ± standard deviation across all five runs.

Performance evaluation employed multiple complementary metrics to capture different aspects of segmentation quality: Dice coefficient, Sensitivity, Specificity, Accuracy, and Hausdorff Distance (HD). The Dice coefficient quantifies spatial overlap between predicted and ground-truth masks, while sensitivity and specificity reflect clinical detection performance for positive and negative regions, respectively. Accuracy provides an overall measure of pixel-level correctness, and the Hausdorff Distance evaluates the maximum boundary discrepancy, which is particularly important in assessing segmentation precision along lesion margins.

All experiments were conducted under identical training conditions, preprocessing pipelines, and evaluation metrics. This strict methodological uniformity ensures that performance differences between models reflect architectural capabilities rather than implementation bias,

allowing for an objective and reproducible benchmarking analysis. SAMed uses SAM pre-training, and all other architectures are trained from scratch under the unified protocol.

## 3. Results and Analysis

### 3.1. Overall Benchmark Performance

Table 1 summarises the performance of eleven segmentation architectures on the curated sonomammogram dataset. The models cover convolutional, hybrid, transformer-based, and state-space designs, all trained and evaluated under the same protocol.

Table 1: Benchmarking of 11 segmentation models on the sonomammogram dataset.

| Model | Dice | Sensitivity | Specificity | Accuracy | HD |
|---|---|---|---|---|---|
| nnUNet | $0.836_{\pm 0.171}$ | $0.855_{\pm 0.167}$ | $0.987_{\pm 0.035}$ | $0.972_{\pm 0.050}$ | $4.814_{\pm 6.967}$ |
| nnUNet ResEnc M | $0.834_{\pm 0.176}$ | $0.845_{\pm 0.178}$ | $0.987_{\pm 0.041}$ | $0.971_{\pm 0.054}$ | $4.581_{\pm 6.709}$ |
| nnUNet ResEnc L | $0.837_{\pm 0.169}$ | $0.853_{\pm 0.169}$ | $0.990_{\pm 0.023}$ | $0.973_{\pm 0.047}$ | $4.453_{\pm 6.224}$ |
| nnUNet ResEnc XL | $0.838_{\pm 0.167}$ | $0.855_{\pm 0.163}$ | $0.987_{\pm 0.040}$ | $0.972_{\pm 0.054}$ | $4.571_{\pm 6.508}$ |
| LightMUNet | $0.798_{\pm 0.210}$ | $0.824_{\pm 0.219}$ | $0.983_{\pm 0.052}$ | $0.966_{\pm 0.063}$ | $5.633_{\pm 7.259}$ |
| SegResNet | $0.818_{\pm 0.176}$ | $0.838_{\pm 0.178}$ | $0.984_{\pm 0.034}$ | $0.969_{\pm 0.049}$ | $6.433_{\pm 8.508}$ |
| SwinUNETR | $0.822_{\pm 0.189}$ | $0.845_{\pm 0.184}$ | $0.983_{\pm 0.051}$ | $0.968_{\pm 0.062}$ | $6.032_{\pm 8.631}$ |
| UMambaBot | $0.830_{\pm 0.187}$ | $0.850_{\pm 0.182}$ | $0.985_{\pm 0.042}$ | $0.971_{\pm 0.054}$ | $4.886_{\pm 7.394}$ |
| UMambaEnc | $0.840_{\pm 0.162}$ | $0.853_{\pm 0.176}$ | $0.989_{\pm 0.025}$ | $0.974_{\pm 0.046}$ | $4.753_{\pm 6.499}$ |
| SAMed (p256) | $0.797_{\pm 0.188}$ | $0.857_{\pm 0.177}$ | $0.982_{\pm 0.040}$ | $0.966_{\pm 0.052}$ | $8.512_{\pm 9.922}$ |
| SAMed (p512) | $0.860_{\pm 0.141}$ | $0.875_{\pm 0.150}$ | $0.990_{\pm 0.023}$ | $0.978_{\pm 0.040}$ | $3.896_{\pm 5.472}$ |

Among all evaluated models, SAMed with patch size 512 achieves the highest mean Dice coefficient ($0.860 \pm 0.141$) and one of the lowest Hausdorff Distances ($3.896 \pm 5.472$). Statistical testing confirms that SAMed (p512) yields a significant improvement over nnUNet in Dice, sensitivity, accuracy, and Hausdorff Distance after Holm correction ($p < 0.05$), indicating that large-patch transformer-based modelling is particularly effective for capturing both lesion extent and boundary geometry in breast ultrasound. The significantly higher sensitivity of SAMed (p512) further suggests improved detection of low-contrast or partially obscured lesions, while maintaining very high specificity, demonstrating a strong balance between detection and false-positive control.

The nnUNet family forms a strong convolutional baseline, with Dice scores clustered tightly between 0.834 and 0.838 across all residual encoder variants. Statistical analysis shows no significant differences between nnUNet and any of its residual encoder variants across all evaluated metrics after correction, confirming that residual depth has only a modest effect under the current training regime. These results reinforce that a well-configured CNN architecture remains highly competitive even when compared with more recent hybrid and transformer-based designs.

The U-Mamba models reach Dice scores of 0.830 (UMambaBot) and 0.840 (UMambaEnc), positioning them between the best-performing transformer models and the nnUNet variants. However, no statistically significant differences were observed between either U-Mamba variant and nnUNet across Dice or accuracy after correction, indicating that their

apparent improvements remain within the range of statistical uncertainty. UMambaEnc, which incorporates Mamba blocks throughout the encoder, shows slightly higher mean Dice than UMambaBot and most convolutional models, while maintaining boundary accuracy comparable to nnUNet. This suggests that state-space modelling can complement convolutional features for ultrasound segmentation, although it does not close the performance gap to SAMed (p512) on this dataset.

SegResNet, LightMUNet, and SwinUNETR exhibit intermediate performance. LightMUNet and SegResNet demonstrate statistically significant differences in Dice relative to nnUNet after correction, while SwinUNETR does not. For boundary accuracy, LightMUNet, SegResNet, and SwinUNETR all achieve significantly lower Hausdorff Distances than nnUNet, indicating improved contour precision despite moderate Dice values. LightMUNet trades representational power for efficiency, yielding a lower mean Dice (0.798) with increased variance, consistent with its reduced architectural capacity.

Specificity is uniformly high across all architectures, exceeding 0.98 in every case, indicating consistently reliable background identification. In contrast, sensitivity varies more markedly (0.824–0.875), reflecting different capabilities for detecting small or low-contrast lesions. Only SAMed (p512) demonstrates a statistically significant improvement in sensitivity over nnUNet after correction. The combined evidence of high Dice, low Hausdorff Distance, and statistically significant gains across multiple metrics for SAMed (p512) indicates that its performance improvements reflect genuine boundary fidelity and detection enhancement, rather than merely increased region overlap.

Figure 1 provides qualitative examples across five representative cases. The nnUNet variants generally produce anatomically plausible segmentations, particularly in lesions with clear hypoechoic cores. LightMUNet and SegResNet tend to undersegment small or poorly contrasted lesions and sometimes oversmooth irregular boundaries, consistent with their lower Dice and higher Hausdorff values.

UMambaBot and UMambaEnc deliver coherent masks that preserve lesion morphology with fewer isolated artefacts, supporting the quantitative observation that sequence style modelling helps maintain global consistency. For SAMed, the patch size has a visible effect. SAMed (p256) occasionally overextends lesion boundaries or introduces small spurious regions, while SAMed (p512) yields tighter contours and better alignment with expert annotations. In some well-defined cases, classic nnUNet matches or slightly exceeds SAMed (p256), illustrating that strong convolutional baselines remain relevant when the lesion boundary is sharp and local context is sufficient.

Across all models, failure cases are concentrated in lesions with indistinct, fragmented, or shadowed boundaries. In such cases, both convolutional and transformer-based architectures may misplace the contour or miss fine protrusions. This is consistent with the relatively large standard deviations in Dice and Hausdorff Distance and reflects the intrinsic difficulty of ultrasound interpretation in these scenarios.

## 3.2. Effect of Loss Function

Table 2 reports the impact of different loss functions on nnUNet performance. DiceCE loss achieves the highest mean Dice coefficient (0.836 ± 0.171), together with a balanced trade-off between sensitivity and specificity. Paired statistical testing confirms that DiceCE yields

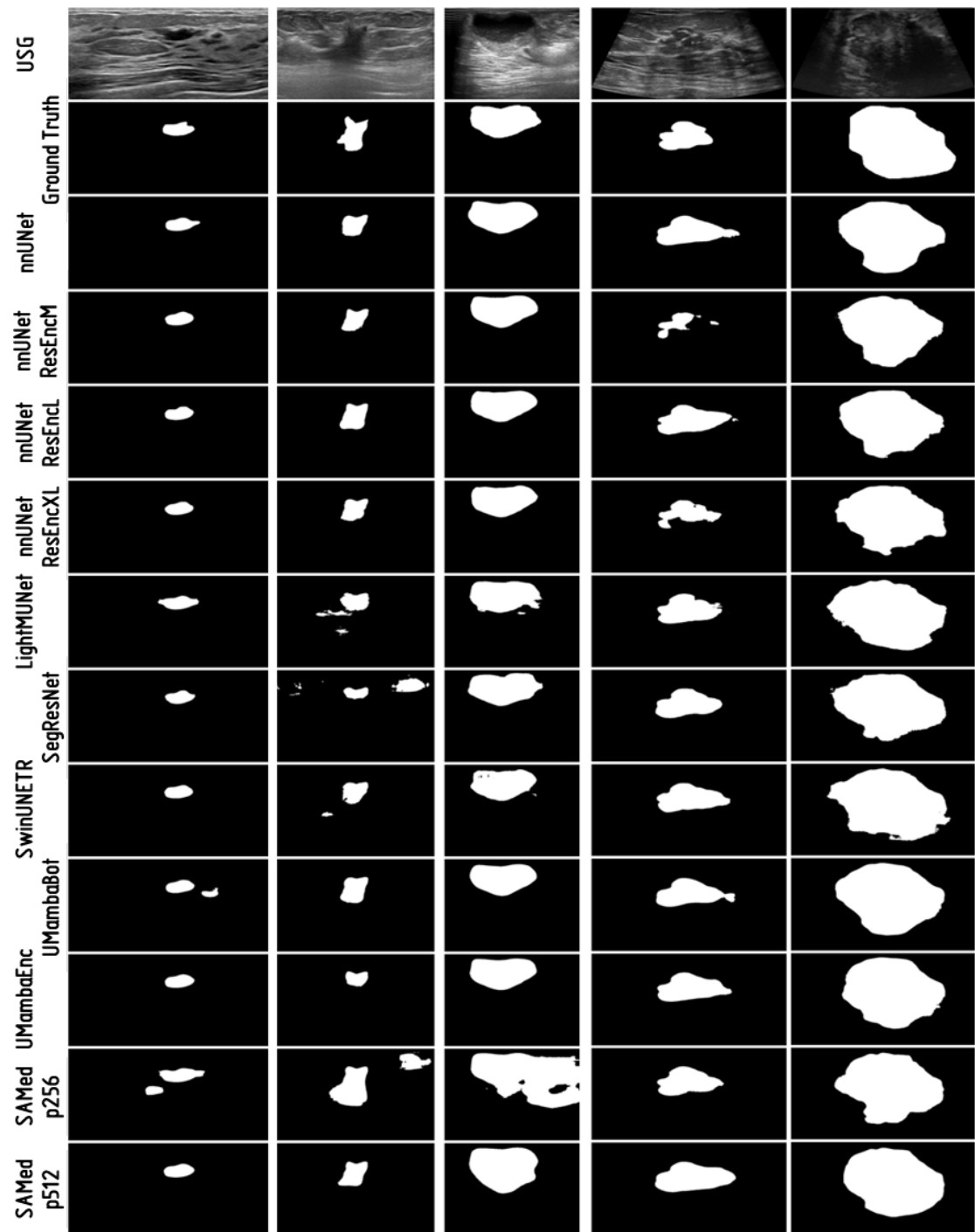

Figure 1: Qualitative comparison of eleven deep learning segmentation models on five representative sonomammograms. The first row shows the ultrasound images, followed by ground truth masks. Subsequent rows display predictions from nnUNet variants, LightMUNet, SegResNet, SwinUNETR, UMamba models, and SAMed (p256/p512).

Table 2: Effect of different loss functions using nnU-Net.

| Loss function | Dice | Sensitivity | Specificity | Accuracy | HD |
|---|---|---|---|---|---|
| DiceTopK10Loss | $0.824_{\pm0.188}$ | $0.835_{\pm0.198}$ | $0.989_{\pm0.028}$ | $0.972_{\pm0.049}$ | $4.790_{\pm6.655}$ |
| CE loss | $0.824_{\pm0.202}$ | $0.824_{\pm0.214}$ | $0.989_{\pm0.033}$ | $0.973_{\pm0.048}$ | $4.127_{\pm5.512}$ |
| Dice loss | $0.820_{\pm0.189}$ | $0.856_{\pm0.183}$ | $0.985_{\pm0.031}$ | $0.971_{\pm0.049}$ | $5.475_{\pm7.482}$ |
| DiceCE loss | $0.836_{\pm0.171}$ | $0.855_{\pm0.167}$ | $0.987_{\pm0.035}$ | $0.972_{\pm0.050}$ | $4.814_{\pm6.967}$ |

a statistically significant improvement in Dice over all other loss functions after correction ($p < 0.05$), indicating that the observed gain is not attributable to random variation.

In contrast, sensitivity differs significantly across loss functions. Both CE and Dice-TopK10 losses produce significantly lower sensitivity compared to DiceCE ($p < 0.05$), indicating a reduced ability to detect low-contrast or small lesion regions. Pure Dice loss achieves sensitivity comparable to DiceCE, but without a corresponding improvement in boundary accuracy. These findings suggest that DiceCE provides a more reliable balance between foreground detection and false-negative control.

No statistically significant differences are observed in specificity or accuracy among any of the loss functions, with all methods exceeding 0.98 in specificity and exhibiting similar overall classification accuracy. This indicates that background discrimination remains robust and largely insensitive to the choice of loss function.

For boundary accuracy, however, cross-entropy loss achieves a significantly lower Hausdorff Distance than DiceCE and the other losses ($p < 0.05$), demonstrating superior contour precision and more conservative boundary delineation. This behavior is consistent with the voxel-wise nature of cross-entropy, which penalizes boundary misclassification more uniformly than overlap-based losses.

Overall, these results indicate that DiceCE is preferable when balanced lesion detection and stable region segmentation are required, as it significantly improves sensitivity without sacrificing specificity or accuracy. Cross-entropy loss, while inferior in sensitivity, is more effective for strict boundary regularization, as reflected by its significantly lower Hausdorff Distance. This highlights a fundamental trade-off between detection performance and boundary precision when selecting the training objective for breast ultrasound segmentation.

### 3.3. Impact of number of Training Data

To assess data dependence, nnUNet was trained with increasing fractions of the available training set. Table 3 summarises the results.

Performance improves strongly as the training fraction increases from 5% to 50%, with Dice rising from 0.661 to 0.800 and Hausdorff Distance decreasing from 15.37 to 6.35. Statistical testing confirms that Dice, sensitivity, and Hausdorff Distance differ significantly across all training fractions ($p < 0.05$), demonstrating a clear dependence of both region overlap and boundary accuracy on the amount of training data.

Gains between 50% and 80% are smaller, and performance begins to stabilise as the full dataset is approached. At 100% of the training data, Dice reaches $0.836 \pm 0.171$ and

Table 3: Effect of the number of training images on segmentation performance for nnUnet.

| # Training (%) | Dice | Sensitivity | Specificity | Accuracy | HD |
|---|---|---|---|---|---|
| 5 | $0.661_{\pm 0.318}$ | $0.669_{\pm 0.323}$ | $0.969_{\pm 0.074}$ | $0.945_{\pm 0.084}$ | $15.368_{\pm 16.534}$ |
| 20 | $0.786_{\pm 0.221}$ | $0.817_{\pm 0.225}$ | $0.986_{\pm 0.028}$ | $0.968_{\pm 0.049}$ | $10.241_{\pm 12.256}$ |
| 50 | $0.800_{\pm 0.232}$ | $0.822_{\pm 0.234}$ | $0.982_{\pm 0.059}$ | $0.967_{\pm 0.067}$ | $6.354_{\pm 5.339}$ |
| 80 | $0.813_{\pm 0.214}$ | $0.839_{\pm 0.209}$ | $0.985_{\pm 0.042}$ | $0.970_{\pm 0.051}$ | $5.051_{\pm 7.340}$ |
| 100 | $0.836_{\pm 0.171}$ | $0.855_{\pm 0.167}$ | $0.987_{\pm 0.035}$ | $0.972_{\pm 0.050}$ | $4.814_{\pm 6.967}$ |

Hausdorff Distance reduces to $4.814 \pm 6.967$, indicating further refinement in segmentation quality. The continued statistically significant reduction in Hausdorff Distance with increasing data volume highlights that boundary precision benefits particularly from additional training samples, even when region-level overlap improvements begin to plateau.

For accuracy and specificity, statistically significant differences are observed only between the extreme low-data regime (5%) and the higher training fractions, while no significant differences are detected among the 20%, 50%, 80%, and 100% settings. This indicates that background classification remains highly robust once a minimal amount of training data is available, whereas lesion detection performance continues to improve with increased data exposure.

Overall, these results indicate that, for this architecture and dataset, approximately half of the available training images are sufficient to reach the main performance regime, with additional data primarily contributing to improved boundary localisation and reduced variability rather than substantial gains in Dice. The pronounced and statistically consistent decrease in Hausdorff Distance with increasing data volume suggests that exposure to a wider diversity of lesion shapes, sizes, and echotextures is particularly important for accurate contour delineation in breast ultrasound segmentation.

### 3.4. Comparative Insights

Taken together, the experiments support several observations. First, architectures that explicitly model broader context, such as SAMed and the U Mamba variants, tend to achieve better combinations of Dice, sensitivity, and Hausdorff Distance than purely convolutional baselines. This indicates that access to long-range information is useful in ultrasound, where lesion boundaries are often defined by subtle intensity transitions and extended contextual cues rather than sharp edges.

Second, model robustness does not simply track parameter count. The nnUNet configurations, which are relatively compact and rely on data-driven configuration, remain among the most reliable performers. Their stable behaviour across loss functions and training set sizes supports their role as strong baselines for future studies.

Third, Hausdorff Distance provides complementary information to region-based metrics. Models such as SegResNet and SwinUNETR can achieve reasonable Dice scores while still exhibiting higher boundary errors, which could be clinically relevant in tasks where small deviations affect downstream measurements. Reporting both overlap and distance metrics is therefore important for a complete assessment of segmentation quality.

Finally, the relatively large standard deviations, particularly for Dice and Hausdorff Distance, reflect the heterogeneity of breast lesions in size, shape, and echotexture. This variability exposes limitations of current architectures and highlights the continued need for methods that can better handle very small, low-contrast, or highly irregular lesions.

Overall, the benchmark shows that large patch transformer-based architectures like SAMed (p512) currently offer the best trade-off between lesion detection and boundary precision on this curated dataset. At the same time, nnUNet remains a strong and accessible reference method, and the data scaling and loss function analyses provide practical guidance for configuring future models on breast ultrasound segmentation tasks.

## 4. Discussion

The benchmarking analysis provides several insights into deep learning based segmentation in sonomammography and clarifies how architectural design, data integrity, and the characteristics of ultrasound imaging interact to shape performance.

A consistent pattern across experiments is the importance of context modelling. Architectures that explicitly capture longer range spatial relationships, such as transformer based models (SAMed, SwinUNETR) and state space designs (UMambaEnc, UMambaBot), tend to achieve stronger overall performance than the simpler convolutional baselines. Breast ultrasound images often exhibit low contrast at lesion boundaries and substantial speckle noise, which limits the discriminative power of purely local filters. Mechanisms that aggregate information over larger regions appear to help resolve ambiguous contours and stabilise segmentation in heterogeneous tissue.

At the same time, the nnUNet framework remains a robust and competitive baseline. Across all experiments, nnUNet variants achieve performance close to the best models while relying solely on convolutional operations and automatic configuration. This supports the view that careful preprocessing, normalisation, and optimisation can offset some of the advantages of more elaborate architectures. It also reinforces nnUNet as an appropriate reference point for future breast ultrasound studies, particularly in settings where computational resources or implementation complexity are constrained.

Within the context aware models, SAMed with a patch size of 512 pixels achieves the best trade off between Dice, sensitivity, and Hausdorff Distance. Its large receptive field and attention based encoding likely facilitate the integration of both global context and local texture cues, which is consistent with its improved boundary localisation. The UMamba variants occupy an intermediate position. Their state space layers provide a form of dynamic context propagation that yields performance above most convolutional baselines, but they do not consistently match SAMed on the current dataset. These results suggest that designing architectures that combine efficient local processing with flexible context handling is beneficial for ultrasound segmentation, but no single family is uniformly dominant.

The data volume experiments highlight that both the quantity and diversity of training data are important. Performance improves sharply when the training fraction increases from 5 to 50 percent of the available images and then gradually plateaus as the full dataset is used. The associated reduction in Hausdorff Distance indicates that additional data particularly benefits boundary precision rather than coarse lesion detection. This pattern implies that, once a representative range of lesion appearances and acquisition conditions is

seen during training, further gains from simply adding more similar images may be limited. Future improvements are therefore likely to depend on complementary strategies, such as incorporating data from other centres and devices, more targeted augmentation, or explicitly modelling uncertainty in challenging regions.

A key aspect of this study is the use of an integrity checked dataset with expert verified annotations and patient level splits. Previous work has frequently relied on public breast ultrasound datasets that contain duplicates, label inconsistencies, and non breast images (Pawlowska et al., 2023). Under those conditions, random splitting can lead to information leakage and optimistic estimates of performance. In contrast, the results reported here are obtained under stricter data curation and controlled evaluation. The absolute performance levels are generally lower than some values reported in earlier literature, but they are more consistent across models and better aligned with the known difficulty of ultrasound interpretation. This supports the view that rigorous dataset validation and transparent benchmarking are essential for obtaining reliable estimates of model capability.

The comparative findings also have implications for future model development. Purely convolutional pipelines appear to be approaching a practical performance ceiling on this task, whereas architectures that incorporate broader context, such as transformers and state space models, offer measurable gains at the cost of increased complexity. An attractive direction is therefore the design of hybrid models that retain the efficiency and inductive biases of convolutions while adding lightweight mechanisms for non local information flow. In parallel, extending evaluations beyond point estimates to include calibration, uncertainty measures, and case wise error analysis will be important for understanding how these systems behave in clinical scenarios.

### 4.1. Clinical Implications

Accurate segmentation of breast lesions in ultrasound directly affects diagnostic assessment and treatment planning. Reliable boundary delineation supports consistent estimation of lesion size, shape, and margin characteristics, which are integral to BI-RADS categorisation, malignancy risk assessment, and longitudinal follow up. The observed reductions in boundary error for the best performing context aware models suggest that they could improve the precision of these measurements, particularly in dense or heterogeneous breast tissue where manual delineation is challenging.

The curated dataset and patient level partitioning used in this work provide a foundation for reproducible evaluation, which is a prerequisite for clinical translation. Benchmarks that are based on validated data and explicit protocols help ensure that reported segmentation accuracy reflects genuine clinical capability rather than artefacts of dataset construction. By quantifying how architecture, loss design, and data volume influence robustness, the present study offers practical guidance for integrating segmentation modules into computer assisted diagnosis pipelines and decision support systems.

### 4.2. Limitations and Future Directions

Despite its scope, this study has several limitations. First, although the dataset is multi institutional, all images originate from centres in a single country, and the cohort may not fully represent the diversity of breast anatomy and imaging protocols encountered interna-

tionally. External validation on datasets from other regions and devices will be necessary to assess generalisability.

Second, all models were trained and evaluated under a common set of hyperparameters and training schedules. This design supports a fair comparison but may not reflect the optimal configuration for each individual architecture. It is possible that some models could achieve higher performance with more extensive tuning or task specific modifications.

Third, the data consist of two dimensional static ultrasound images with single expert consensus segmentations. Temporal information, three dimensional acquisitions, and explicit modelling of inter observer variability were not considered. These aspects could be important in clinical practice and represent natural extensions of the current benchmark.

Finally, the dataset, although carefully curated, is of moderate size by modern deep learning standards, and access constraints may limit direct reuse in some settings. Future work will focus on extending the benchmark to larger and more diverse cohorts, incorporating multi centre external validation, and exploring uncertainty aware and clinically guided evaluation criteria. Addressing these points will be critical for translating segmentation models into robust and trustworthy tools for routine breast imaging.

### 4.3. Dataset Availability

Part of the dataset is published in (Pawłowska et al., 2024), and for any other questions related to the dataset, contact N.Z.

## 5. Conclusion

This study delivers a comprehensive, methodologically consistent benchmark of deep learning architectures for breast ultrasound segmentation. Transformer- and state-space models, such as SAMed and U-MambaEnc, achieved the highest boundary accuracy and contextual understanding, while classical CNNs like nnU-Net remained competitive when rigorously configured. This demonstrates that segmentation success in ultrasound depends less on architectural novelty and more on data fidelity, reproducibility, and balanced optimisation.

Beyond model performance, this work represents a methodological correction for the field of ultrasound-based breast cancer imaging. It exposes how unreliable datasets have distorted prior results and provides a transparent foundation upon which future methods can be developed and compared without bias.

Looking forward, the integration of multi-centre, multi-device datasets, combined with hybrid architectures that balance interpretability and contextual depth, will be essential to advancing the clinical readiness of deep learning–based segmentation systems. The benchmark presented here not only quantifies the state of the art but also redefines the standards for credibility, reproducibility, and clinical relevance in breast ultrasound image analysis.

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
