# OpenReview forum: "A Comprehensive Benchmarking and Systematic Analysis of Deep Learning Models for Sonomammogram Segmentation"
_MIDL.io/2026/Validation_Papers — MIDL 2026 - Validation Papers Poster_

### Official Review · Reviewer_e5Zn · 2025-12-22

**Confidence:** 4
**Preliminary Rating:** 4

**Summary:**

This paper presents a comprehensive and methodologically consistent benchmark of deep learning models for breast lesion segmentation in ultrasound images. The authors introduce a carefully curated multi-centre dataset with expert-verified annotations and strict patient-level splits, addressing known integrity issues in widely used public datasets. Eleven representative architectures spanning CNNs, transformers, and state-space models are evaluated under a unified training and evaluation protocol using multiple clinically relevant metrics. The experiments systematically analyse architectural performance, loss function choice, and the effect of training data volume. Overall, the work provides a reliable reference point for future research and highlights the importance of data quality and reproducibility over architectural novelty.

**Strengths:**

The main strength of the paper lies in its rigorous benchmarking methodology and strong emphasis on data integrity. The curated dataset, expert consensus annotations, and patient-level splitting directly address critical shortcomings of prior ultrasound benchmarks and substantially improve the credibility of the reported results. The evaluation is comprehensive, covering a broad range of modern architectures under identical experimental conditions, which enables meaningful and fair comparison. The inclusion of multiple metrics, particularly Hausdorff Distance alongside Dice, provides a more clinically relevant assessment of segmentation quality. The paper is clearly written, well-structured, and well grounded in prior work, making it a valuable resource for the medical imaging community even though it does not propose a new model.

**Weaknesses:**

The study is limited to two-dimensional static ultrasound images from centres within a single country, which raises questions about generalisability to other populations, devices, and acquisition protocols. While the unified training setup ensures fairness, it may disadvantage certain architectures that typically require task-specific tuning to reach their full potential. The dataset size, although carefully curated, remains moderate by current deep learning standards, which may partially explain the large performance variance observed across cases. In addition, the analysis focuses primarily on segmentation accuracy and boundary error, without considering uncertainty estimation, calibration, or inter-observer variability, which are relevant for clinical deployment. These limitations are acknowledged by the authors but still constrain the broader impact of the findings.

**Detailed Comments:**

The paper would benefit from a brief discussion on computational cost or training efficiency, particularly for transformer-based models such as SAMed, to help contextualise performance gains. Clarifying whether the curated dataset will be publicly released, and under what conditions, would increase the practical value of the benchmark. A short case-wise error analysis highlighting typical failure modes could further strengthen the clinical interpretation of the results.

**Justification Of The Preliminary Rating:**

This paper makes a solid and timely contribution as a validation and benchmarking study rather than a method paper. Its primary value lies in correcting methodological issues that have affected prior work, especially dataset integrity and inconsistent evaluation protocols. While the work does not introduce a novel model and is limited in geographic diversity and data scale, the experimental design is rigorous, the analysis is thorough, and the conclusions are well supported by the results. The benchmark provides meaningful guidance to the community and establishes a reliable reference for future developments, which justifies acceptance despite the acknowledged limitations.

**Questions To Address In The Rebuttal:**

How well do the authors expect the benchmark results, particularly for SAMed and U-Mamba models, to generalise to ultrasound data from different countries, vendors, or acquisition protocols?
Do the authors anticipate that model rankings would change substantially with architecture-specific hyperparameter tuning, or are the observed trends likely to remain stable?

---

### Official Review · Reviewer_xB3W · 2025-12-25

**Confidence:** 4
**Preliminary Rating:** 4
**Final Rating:** 4

**Summary:**

This paper addresses the issue of unreliable benchmarking in breast ultrasound segmentation, which stems from flaws in existing public datasets (e.g., duplicates and incorrect labels). The authors introduce a carefully curated multi-centre dataset comprising 3,494 images from 1,727 patients, featuring expert-verified annotations and strict patient-level splitting. Leveraging this high-quality dataset, the authors conduct a systematic benchmark of 11 deep learning architectures—ranging from standard CNNs to Transformers and State-Space Models (e.g., SwinUNETR, SAMed, U-Mamba)—using a unified training and evaluation pipeline. Furthermore, the study includes an analysis of loss functions and training data volume.

**Strengths:**

1.The authors provide a critical analysis of the limitations in existing datasets and subsequently collect and annotate a new dataset using rigorous standards.
2.The study selects a comprehensive set of baseline models and performs a fair comparison using an identical data processing pipeline and training procedure.
3.The authors conduct valuable experimental analyses on the impact of different loss functions and variations in training data volume.

**Weaknesses:**

1.Although the dataset is multi-centre, all centres are located in the same country. While the authors acknowledge this limitation, for a "Comprehensive Benchmarking" paper in the Validation Track, the lack of external validation on datasets from different populations or device manufacturers limits the claims regarding generalizability.
2. The dataset is not yet publicly available.
3. The best-performing model is SAMed, which relies on the SAM architecture and typically utilizes massive pre-training (SA-1B). The paper does not explicitly discuss how much of the performance gap is due to the architecture itself versus the scale of pre-training (compared to models trained from scratch, like nnU-Net, or those using standard ImageNet weights). This conflation makes it difficult to isolate "architectural superiority." The usage of pre-trained weights for the other comparison models also needs to be clarified.

**Detailed Comments:**

1. Please clarify the distribution of image resolutions prior to resizing, as well as the distribution of acquisition device types used in the dataset.
2. I would appreciate seeing some visualization of failure cases to gain a more detailed understanding of the specific limitations of each model.

**Justification Of Final Rating:**

I would like to thank the authors for their detailed and constructive response. The rebuttal has satisfactorily addressed my concerns regarding dataset availability (accessible upon request), the diversity of acquisition devices.

The greatest contribution of this article is the dataset; the authors should clearly specify the data access agreement in the article.

Given the rigorous data curation and the systematic evaluation protocol, I maintain my positive rating for this work.

**Justification Of The Preliminary Rating:**

The paper is a strong candidate for the Validation Studies Track due to its rigorous approach to data curation and the establishment of a unified evaluation pipeline. The authors correctly identify and address the critical issues of label noise and duplication in existing public datasets, which is a valuable contribution to the community. The systematic analysis of loss functions and data scaling further adds to the paper's practical utility. However, the preliminary rating is constrained by some major concerns that affect the impact and validity of the benchmark.  I expect the authors to address these issues in their subsequent rebuttal.

**Questions To Address In The Rebuttal:**

1.Will the data be made publicly available?
2. Please clarify the specific usage of pre-trained weights for all models.
3. Can you provide a clinical usability analysis, specifically regarding inference hardware requirements, inference speed, and an analysis of edge cases?

---

### Official Review · Reviewer_jefS · 2026-01-07

**Confidence:** 4
**Preliminary Rating:** 4

**Summary:**

This paper addresses issues such as duplication and labeling errors in existing breast ultrasound datasets. A multi-center curated dataset containing 3,494 images (from 1,727 patients) was constructed, and 11 representative deep learning architectures were evaluated using a unified benchmark protocol. The analysis of model performance and the impact of loss functions and training data volume was conducted through five metrics, including the Dice coefficient and sensitivity. This benchmark provides a reproducible reference for breast ultrasound segmentation.

**Strengths:**

1. The authors provided a high-quality curated dataset, and addressed a critical gap in existing breast ultrasound datasets by eliminating duplicates, mislabeled cases, and non-breast images.
2.  Authors evaluated 11 representative architectures spanning CNNs, transformers, and state-space models (e.g., nnU-Net variants, SwinUNETR, U-Mamba, SAMed) under identical preprocessing, training, and evaluation settings. This standardization eliminates implementation bias and enables objective comparison!
3. The benchmark directly informs clinical translation by highlighting models (e.g., SAMed p512) that balance lesion detection (sensitivity) and boundary accuracy (Hausdorff Distance), which are key factors for BI-RADS categorization and treatment planning.

**Weaknesses:**

1. Despite being multi-institutional, all data originate from a single country (Poland), raising concerns about how well results transfer to diverse global populations, breast anatomies, and imaging protocols. This lack of geographic and demographic diversity severely limits the benchmark’s external validity.
2. The dataset consists solely of 2D static images with single-consensus annotations. It ignores critical clinical dimensions: 3D ultrasound data, temporal dynamics (e.g., real-time imaging), and inter-observer variability (a major challenge in clinical segmentation). By excluding these factors, the benchmark does not reflect the full complexity of real-world ultrasound interpretation.
3. The study compares entire architectures but provides little insight into why certain models perform better. For example, SAMed p512’s superiority is attributed to 'large-patch transformer-based modelling,' but there is no ablation of patch size (e.g., 128 vs. 256 vs. 512) or attention mechanism variants.

**Detailed Comments:**

No further comments, thanks for authors.

**Justification Of The Preliminary Rating:**

This submission presents a valuable contribution to the field of breast ultrasound segmentation by addressing long-standing gaps in dataset quality and benchmark standardization, two critical barriers to reproducible research in medical image analysis. However, several non-fatal but substantial limitations prevent a stronger rating, leading to the preliminary decision of Weak Accept.

**Questions To Address In The Rebuttal:**

1. For each architecture, what hyperparameters were tuned (if any) and what search space was used?
2. For SAMed, provide ablation results showing how patch size, fine-tuning strategy, and attention mechanism design impact performance. For U-Mamba models, isolate the contribution of Mamba blocks (e.g., replace Mamba layers with CNNs and re-evaluate) to clarify whether state-space modeling provides unique benefits beyond convolutional features.

---

### Author Rebuttal · Authors · 2026-01-25

We thank all reviewers for their careful reading of the manuscript and for their constructive and insightful feedback, particularly regarding the dataset integrity, the rigour of the unified benchmarking protocol, and the breadth of evaluated architectures.

We have addressed all comments and questions to the best of our knowledge and added new analyses, ablation studies, and clarifications where requested. Despite the limited revision time, we conducted additional experiments to directly respond to the reviewers’ concerns and to improve the methodological transparency of the study.

We hope that the revisions satisfactorily address the raised concerns and lead to a favourable outcome.

---

### Meta-Review · Area_Chair_YdYW · 2026-02-08

**Recommendation:** Accept (Oral)
**Confidence:** 4

**Metareview:**

This paper presents a carefully curated breast ultrasound dataset and a rigorous, unified benchmark of 11 segmentation architectures. Reviewers valued the strong focus on data integrity, fair evaluation, and clinical relevance.
Concerns about limited geographic diversity, dataset availability, and architectural interpretability were raised. The authors addressed these through clear scope framing, explicit data-access clarification, transparency on pretraining, and added ablation studies explaining performance differences.
The dataset used in this study is not yet publicly available. The authors state that it will be shared upon reasonable request for research purposes, subject to ethical and institutional constraints. Given that the curated dataset is a central contribution of this work, it is important that this access commitment is clearly stated and upheld. I encourage the authors to explicitly specify the data access conditions in the final version and to follow through on making the dataset available as promised, as this will be essential for reproducibility and long-term impact.

---

### Decision · Program_Chairs · 2026-02-14

Accept (Poster)